# Mental Stress in Medical Students during the Pandemic and Their Relation to Digital and Hybrid Semester—Cross-Sectional Data from Three Recruitment Waves in Germany

**DOI:** 10.3390/ijerph191711098

**Published:** 2022-09-05

**Authors:** Arndt Büssing, Michaela Zupanic, Jan P. Ehlers, Angelika Taetz-Harrer

**Affiliations:** 1Quality of Life, Spirituality and Coping, Faculty of Health, Witten/Herdecke University, 58313 Herdecke, Germany; 2Interprofessional and Collaborative Didactics in Medical and Health Professions, Witten/Herdecke University, 58455 Witten, Germany; 3Didactics and Educational Research in Health Care, Witten/Herdecke University, 58455 Witten, Germany; 4Office of Student Affairs in Human Medicine, Witten/Herdecke University, 58455 Witten, Germany

**Keywords:** stress perception, wellbeing, cool down, medical students, COVID-19 pandemic, digital semester, teaching formats

## Abstract

Background: Because of the COVID-19 pandemic, students had to interrupt their regular studies, and universities changed their teaching formats. The aim of this study was to analyze medical students’ stress perception, wellbeing, life and work satisfaction, and cool down reactions, and to compare the survey data of online and hybrid semesters with pre-pandemic education formats in-person. Methods: Cross-sectional surveys at three time points enrolling 1061 medical students (58% women; 24.4 ± 3.4 years); 30.8% from pre-pandemic formats in-person, 22.8% from pandemic online semesters, and 46.1% from pandemic hybrid semesters. Results: Both students’ stress perception and psychological wellbeing decreased during the pandemic semesters. Their satisfaction with the university support was at its lowest during the hybrid semesters. Regression analyses indicated that students’ stress perception can be explained only to some extent by their general dissatisfaction with their medical studies or teaching formats. Conclusions: The lockdowns affected students in more ways than simply their teaching formats. Students require individual support to adjust to difficult situations, and particularly medical students in their preclinical phase compared to students in their clinical phases. These are challenges for the medical education system, which must find ways to be prepared for future times of crisis and insecurity.

## 1. Introduction

During the first phases of the COVID-19 pandemic, people reported phases of insecurity and anxiety, and most were in fear of becoming infected and having a complicated course of disease [1,2,3]. The lockdowns with their social restrictions further burdened people in different ways. A systematic review revealed an increased prevalence of major depressive disorders (28%) and anxiety disorders (26%) in the population of different countries, particularly in women and younger people [4]. In Germany, we found a further decline in people’s wellbeing, particularly with the onset of the second lockdown during the winter of 2020 [5], which was perceived more strongly by women and younger people [6]. While it is true that the pandemic-related social restrictions were perceived more strongly by young people, what about medical students who want to acquire medical knowledge and skills during their medical studies in order to later become medical doctors and help sick people?

During the first phase of the pandemic in 2020, medical schools were in lockdown like all others, and paused clinical internships and in-class training, later offering online education programs [7,8,9]. Thus, students had to consider how to deal with this challenging situation. Later on during the pandemic, all courses were digitally administered (“online”) and then 1.5 years later they were applied in a hybrid format with education both online and in-person. As the restrictions have affected them in general, with a decrease of mood states and an increase in study uncertainty, we were interested whether these study formats (online or hybrid during the pandemic compared to in-person formats prior to the pandemic) could have contributed to improving medical students’ wellbeing and reducing their stress perception. For that purpose, we analyzed data from medical students recruited in a German university.

### 1.1. Students’ Stress Perception and Wellbeing during the Pandemic

Stress perception is dependent not only on the stressor, but also on personality factors and coping strategies. It is therefore to be expected that the pandemic related stressors were not perceived similarly in women and men, during the different phases of the pandemic, and in different cultures and societies, etc.

In students from Australia recruited during the pandemic, their stress levels were moderate, without significant differences between students from the different years of their medical course [10]. Nevertheless, in these Australian students a deterioration of their wellbeing due to the pandemic was reported by 68% [10]. In contrast, Chinese medical students had a prevalence of anxiety disorders by 17% and for depression by 25%, and these mental health afflictions were more often perceived by women than by men [11]. Another study reported a 29% prevalence of anxiety symptoms and a 32% prevalence for depressive symptoms among international medical students in China [12]. There might in fact be cultural differences for pandemic related anxiety perceptions. A systematic review by Liyanage et al. reported a 41% prevalence of anxiety in university students [13]. Subgroup analyses revealed that the prevalence in Asia was 33%, while it was 51% in Europe and 56% in the USA [13]. Among medical students from Turkey, one third reported that their sleep and appetite were more affected by the pandemic than before, and they were worried about infection with the SARS-CoV-2 virus [14]. Medical students from Jordan were “concerned about family members’ affection” and about the “inability to get clinical sessions and labs” during the pandemic, and most suffered from mental disorders [15]. In a study by Guo et al., the reasons for students’ stress level were related to “trust in government institutions” during the pandemic, “delay/availability of standardized exams”, and insecurities about the “impact on rotations/residencies” [16]. In medical students from Saudi Arabia, Meo et al. reported a decline of work performance and time spent studying because of the quarantine [17].

Thus, the pandemic challenged medical students who felt stressed, depressed, and insecure about their further medical education. Nevertheless, a meta-analysis by Puthran et al. reported a 28% prevalence of depressive states in medical students already before the pandemic [18]. In that study, the first year, first semester students were more depressed as compared to students in the later phases of study [18]. A representative online survey in Germany prior to the pandemic revealed that students experience above-average stress, and almost half of them indicated a low stress resilience [19]. An evaluation of students’ medical data showed that 17–22% were diagnosed with a mental illness, predominantly emotional exhaustion/burnout, and depressive mood states, already before the pandemic [20]. These conditions can easily lead to delays and interruptions in their studies and even the dropout of their studies [21].

Nevertheless, several medical students were asked within the first phase of the pandemic to help health care professionals as supplementary assistants [22] and worldwide, students showed solidarity with health care professionals and supported them as volunteers [23,24,25]. In medical students from Germany, the main reasons to work as volunteers were altruistic intentions and the practical application of their acquired knowledge [26]. Nevertheless, health care professionals who were supported by their future colleagues had their own fears and worries. People in the health care system had a high risk of infection with the SARS-CoV-2 virus during the first phase [27], and had symptoms of depression (47%), anxiety (50%) and low quality of life (45%) [28]. 

### 1.2. Support of Students by Their Universities: Adjustment of Leaning Formats 

The situation summarized above indicates that universities have to avoid insecurities because of study interruptions and support their students by adjusting teaching formats. This could contribute to decreasing students’ stress and to increasing their wellbeing, even when the pandemic restrictions are still active. Further, there is a general need for programs to detect students at risk and implement prevention strategies to support them. Several of these stressors and related mental health outcomes were perceived already before the pandemic, which may have further aggravated the inherent stresses within education. 

As we had started to determine the number of our university’s students at risk of stress and burnout (and plan the types of university support services) prior to the COVID-19 pandemic [29], we are now able to compare data from our medical students in the Witten/Herdecke University (Germany) prior to the pandemic and during the pandemic, both in a strict lockdown semester at the start of the pandemic 2020 where all courses were digitally administered (“online”), and also 1.5 years later when courses were applied in a hybrid format with both education online and in-person. This offers the opportunity to directly compare the in-person semesters (with the university’s practical studies and many small learning and working groups [30]) with the digital content of the lockdown semesters, without direct contact with fellow students or patients, and with the hybrid semesters later on. 

### 1.3. Aim of the Study

It was the aim of this analysis to evaluate German students’ stress perception, wellbeing, life and work satisfaction, and their cool down reactions within these specific education formats that were offered to deal with necessary social restrictions because of the pandemic. For that purpose, we compared data from three time-points with different teaching formats: (1) in-person, prior to the pandemic, (2) online teaching because of the stricter lockdown restrictions, (3) hybrid format teaching during the later phases of the pandemic. We expected that the pandemic had affected students’ wellbeing and analyzed the outcome variables also related to the phase of their study, categorized as either younger students in their preclinical phases (semesters 1–4) or older students in the clinical phases of their study (semesters 5–13). Finally, we analyzed the predictors of their wellbeing, stress perception, and work engagement. 

The theoretical basis of our project is a modified Job Demands–Resources Model [31] which assumes two important components (stressors and resources) and two important processes (impairments and motivation) that influence a person’s perception of burden. For this analysis, we focused on the stressors and impairment rather than on their resources and motivations. 

## 2. Materials and Methods 

### 2.1. Description of the Study Sample

For this study from Germany, we relied on data from three waves of recruitment: (1) students from recruiting wave 1 (W1) recruited in the Winter Semester 2018/2019 (following the students’ official progress test) which was given in-person before the pandemic; (2) students from wave 4 (W4) recruited in Summer Semester 2020 at the start of the pandemic and thus studied online; (3) students from wave 7 (W7) recruited in Winter Semester 2021/2022 during the pandemic which was a hybrid semester with both online and presence education. Wave 7 includes more students, as twice as many students have been admitted per semester since the summer semester 2019 and have started a medical degree program at the Witten/Herdecke University. Before the summer semester 2019, 42 students per semester began their medical studies, and since the summer semester 2019, 84 students per semester have begun medical studies.

Medical students from the Witten/Herdecke University, Germany (inclusion criterion), were invited to participate in this anonymous survey via emails from the university’s vice president for Teaching and Learning to all medical students. To avoid pathologizing the students, we did not ask for (suspected) psychiatric diagnoses. Students were reminded of the survey three times at seven-day intervals via email. The study was approved by the ethical commission of Witten/Herdecke University (#132/2017). The W1 questionnaire was applied as paper-pencil version, and all other online via LimeSurvey.

### 2.2. Measures

Apart from basic sociodemographic and semester-related data, students responded to standardized measures that are described in the following:

#### 2.2.1. Stress Perception 

Stress perception was addressed with the 10-item *Perceived Stress Scale* (PSS) [32]. Its internal consistency was moderate in the primary study (Cronbach’s alpha = 0.78) [32], but good in medical students from Germany (Cronbach’s alpha = 0.89) [26]. All items refer to emotions and thoughts and how often one may have felt or thought a certain way within the last month. Example items are, “How often have you felt nervous and “stressed”?” (item 3) or “How often have you found that you could not cope with all the things that you had to do?” (item 6). The scores range from 1 (never) to 4 (very often); higher scores would thus indicate greater stress perception.

#### 2.2.2. Wellbeing

To assess students’ psychological well-being, the *WHO-Five Well-being Index* (WHO-5) was applied [33]. The scale’s internal consistence is good among medical students (Cronbach’s alpha = 0.85) [26]. Representative items are, “I have felt cheerful and in good spirits” (item 1) or “My daily life has been filled with things that interest me” (item 5). Participants assess how often they had the respective feelings within the last two weeks, ranging from 0 (at no time) to 5 (all of the time). WHO-5 scores < 13 may indicate depressive mood states.

#### 2.2.3. Life Satisfaction

Life satisfaction was measured with the *Brief Multidimensional Life Satisfaction Scale* (BMLSS; Cronbach’s alpha = 0.87) [34] in its eight-item version. It addresses intrinsic (myself, life in general), social (friendships, family life), external (medical studies, where I live), and health (ability to manage daily life concerns, health situation) domains. Each of these items was introduced by the sentence, “I would describe my level of satisfaction as …”, and scored on a 7-point scale ranging from dissatisfaction to satisfaction. The mean scores were referred to a 100% level (‘delighted’).

We also used the instrument’s *Support Satisfaction* module which addresses satisfaction with the support from fellow students, support from the university, cohesion among fellow students, and exchange between fellow students. A further independent item addresses satisfaction with the current study term/semester (T1). Each of these items was introduced by the sentence, “I would describe my level of satisfaction as …”, and scored on a 7-point Likert-scale ranging from dissatisfaction to satisfaction. The mean *Support Satisfaction* scores were referred to a 100% level (‘delighted’). 

#### 2.2.4. Work Engagement

Work engagement was measured with the *Utrecht Work Engagement Scale* (UWES) which addresses attitudes of vigor, dedication, and absorption [35]. For this study, we used the 9-item shortened version (UWES-9; Cronbach’s alpha ranging between 0.85 and 0.92) which has similar psychometric properties as the long version. In the introduction of the instrument we stated that the term “work” should be related to their “work for study”. Specific items are, “At my work, I feel strong and vigorous” (item 5) or “I am immersed in my work” (item 8), etc. The items are scored on a 7-point Likert scale, ranging from 0 (never) to 6 (always / every day). 

#### 2.2.5. Cool Down

Stressful situations during the job may result in emotional exhaustion and, as a strategy to cope and to remain ‘functional’ in the job, emotional distancing towards the patients [36]. To measure these cool down reactions, the 9-item *Cool down Index* (CDI; Cronbach’s alpha = 0.84) was applied. This scale was further tested in medical students and showed good internal consistency, too (Cronbach’s alpha = 0.83) [26]. All statements were introduced by the phrase, “When dealing with the people I have to care for or patients I meet during my medical training, I notice that …”. Specific perceptions and experiences are, “Patients’ personal problems and worries often simply become too much for me”; “I often no longer have the patience to listen to them”; “I have to withdraw with increasing frequency to protect myself”. The frequency of perceptions was scored from 1 (a few times a year or less), 2 (once a month or less), 3 (a few times a month), 4 (once a week); 5 (a few times a week) to 6 (every day).

### 2.3. Statistical Analyses

Descriptive statistics are presented as frequencies for categorical variables (%) and as mean ± standard deviation (SD) for numerical variables. Between-group comparisons for categorical variables were performed with Pearson’s Chi^2^ Independence Test, and one-way analysis of variance with repeated measures for numerical variables. Regression analyses were performed stepwise to predict wellbeing, stress performance, and work engagement by gender, phases of study, and five items of the BMLSS (Satisfaction of medical studies; H3, T1–T4). Given the exploratory character of this study, we set a stricter significance level at *p* < 0.001.

## 3. Results

### 3.1. Description of the Study Sample

Within the three recruitment waves, the response rates of medical students differ. At W1, 59% have participated the paper-pencil survey, but we did not reach those students who were already working in the clinics (semesters 10 to 13). During the online semester (W4), most students were at home and thus the response rate was low (36%), while during the hybrid semester (W7) the response rate was much better (62%), including the semesters 10–13. 

The gender proportion within the three waves prior and during the pandemic is similar, and also students’ mean age was similar (Table 1). However, during the online semester (W4) more students from the preclinical semesters participated, while the hybrid semester (W7) had the same proportion of students in their preclinical and clinical phase as prior to the pandemic (W1). Within the preclinical and the clinical semesters, there were no significant differences in gender proportion for W1 and W4, but there was a trend for more women in the preclinical semesters in W7 compared the women in the clinical semesters of W7 (*p* = 0.046). Further, there were no significant differences for students’ mean age in W1 and W4, but a marginal difference in W7 (f: 24.1 ± 2.6 vs. m: 25.0 ± 3.3; *p* = 0.003) (data not shown).

Students’ satisfaction with medical studies was high within the pre-pandemic semester and also within the online semester, while it slightly decreased in the hybrid semester (W1 vs. W7: *p* < 0.001) (Table 1). Similarly, the Support Satisfaction was slightly lower in W7, while the difference in W1 and W4 semesters was not statistically significant (Table 1).

### 3.2. Changes of Stress Perception and Psychological Wellbeing in the Study Sample

Students’ stress perception was significantly lower in the digital semester (W4) compared to the semester before the pandemic (W1), and was higher in the pandemic hybrid semester (W7) compared to the digital semester (W4) (Table 2). These changes were observed both in preclinical and clinical students, while, however, the differences between W4 and W7 were not significantly different in students in their clinical semester.

Students’ wellbeing decreased strongly during the course of the pandemic, particularly in the younger students within their preclinical phases (Table 2). A decrease of psychological wellbeing was observed also in students within their clinical phases, but this decrease was statistically not significant.

Students’ multidimensional life satisfaction was very high prior to the pandemic, and decreased significantly during the pandemic (to still high scores). However, these decreases were not significant in clinical phase students (Table 2). Further, students’ satisfaction with university support decreased significantly within the online semester, particularly in women. However, during the same time students’ satisfaction with the support by fellow students or cohesion among fellow students did not significantly change (Table 2).

Students’ work engagement decreased only slightly, more relevantly in preclinical students, but not significantly in medical students during their clinical semesters (Table 2). With respect to emotional exhaustion and emotional distancing, as measured with the Cool Down Index, no significant changes related to the pandemic were observed (Table 2).

With respect to gender, the decreases of stress perception were observed both in female and male students (Table 3), with no significant differences in W4, while in W7 the stress was perceived less strong by male students (f: 17.7 ± 3.3 vs. m: 15.9 ± 4.0; *p* < 0.0001). Wellbeing was decreasing in W4 (without significant differences in W1 and W4 between women and men), and lower in women compared to men in the hybrid semester W7 (f: 13.1 ± 5.1 vs. m: 14.8 ± 5.0; *p* = 0.003). A significant decrease of life satisfaction and work engagement was observed only in women in the hybrid semester, but not in men (Table 3). A decrease of satisfaction with university support was perceived in the hybrid semester by female students but not by male students (Table 3), while satisfaction with fellow students’ support and cohesion was not perceived differentially by women and men (Table 3).

There were no significant differences for work engagement between women and men in W1 and W4, but a trend for marginally lower scores in W7 women (f: 3.5 ± 1.0 vs. m: 3.8 ± 1.1; *p* = 0.046). For Cool Down, no significant gender-related changes were observed, yet there was a trend for lower scores in W7 women (f: 15.1 ± 7.1 vs. m: 17.2 ± 9.0; *p* = 0.046), while female and male students did not differ for this variable in W1 and W4 (data not shown).

### 3.3. Correlations between Indicators of Stress and Psychological Wellbeing

To assess the intercorrelations of outcome variables, these were correlated in the whole sample of students (Table 4). Interestingly, students’ stress perception was not significantly related to their psychological wellbeing, but was weakly negatively correlated to their life satisfaction and positively to Cool Down. Their life satisfaction was moderately positively correlated with support satisfaction, work engagement, wellbeing and negatively with Cool Down. Work engagement was further moderately positively associated with Support satisfaction. Students’ Cool Down perception was further weakly positively related to stress perception, and inversely to support satisfaction, indicating a buffering effect. While age and semester are correlated (r = 0.305), age was not related with any of the other variables (data not shown); instead, semester was marginally positively related to stress perception and negatively to work engagement, but inversely correlated with semester count (Table 4).

### 3.4. Predictors of Wellbeing, Stress Perception and Work Engagement

To analyze how the main outcome variables (wellbeing, stress perception, and work engagement) can be predicted, we performed stepwise regression analyses. The following variables were included in the stepwise regression models: gender (women/men), phase of the study (preclinical/clinical phase), and five dimensions of satisfaction (medical studies, support from university, support from fellow students, cohesion among fellow students, exchange between students). These were tested in the respective waves of recruitment in terms of the depending variables wellbeing (WHO-5), stress perception (PSS) and work engagement (UWES). In the respective regression models (Table 5, Table 6 and Table 7) only the significantly included variables were depicted.

With respect to wellbeing (Table 5), students’ satisfaction with the medical studies and with the support received by the university were of some relevance in W1, but with low explanatory power (R^2^ = 0.09). During the online semester (W4), satisfaction with the studies and with university’s support was still relevant, but also of being in the clinical phase of the medical studies (explaining together 17% of variance). Students’ wellbeing in the hybrid semester (W7) can be explained by five variables, best by satisfaction with the studies in general, and further by being in the clinical phase of studies, satisfaction with university’s support, fellow students’ support, and by male gender (explaining together 36% of variance).

Students’ stress perception was explained by only a few satisfaction variables, yet with weak predictive power (Table 6). Satisfaction with the studies was a negative predictor of stress perception. In the hybrid semester and also prior to the pandemic, male gender was a negative predictor (Table 6).

With respect to work engagement, students’ satisfaction with the medical studies in general was the best and positive predictor in all waves (Table 7).

## 4. Discussion

The relevant findings of this study were that students’ stress perception decreased during the pandemic semesters, indicating an improvement of stress, while their psychological wellbeing decreased strongly during the pandemic semesters, indicating a worsening of their psychological stability. This finding is surprising. In line with well-being, life satisfaction decreased in relative terms. However, the reported life satisfaction was still at a high level. Surprisingly, students’ satisfaction with the support by the university decreased significantly within the online semester and was lowest during the hybrid semester, while their satisfaction with fellow students’ support and cohesion did not change. This is contrary to our expectation that the efforts of the university to apply hybrid teaching (which allows more social contact) would have improved their psychological wellbeing. This would indicate that during the pandemic, the well-being of students, regardless of the teaching mode, decreased significantly because of the pandemic, while their stress levels can be explained by the change from in-person to online to hybrid semesters. However, depending on the phase of their study (preclinical versus clinical semesters) and gender, these perceptions may differ.

### 4.1. Stress Perception of Students during the Pandemic

In the general population in Germany, with the onset of the pandemic and with the start of the second lockdown, people’s psychological wellbeing deceased significantly [5]. Younger people in particular had low wellbeing and perceived the restrictions more strongly compared to older people [5]. This also applies to medical students who volunteered in the context of the CoronAid initiative during the first phase of the pandemic who also report significantly lower levels of wellbeing at this time (13.5 ± 5.1) [26]. This score corresponds to the findings in this study of German medical students within the pandemic online semester. These perceptions were stronger in students in their preclinical phase compared to students in their clinical phases. It seems that even when the later students were restricted in their medical studies at university, having a focus on the clinical aspects of their studies may have buffered the decrease in their wellbeing to some extent. A further reason to explain this difference might be that the younger students (in their preclinical phase) require social contact when they are confronted with a new surroundings, are not clear about all processes, examinations etc. These concrete social contacts were lacking in the digital semester, while their older counterparts in the clinical semesters have already established these contacts and can rely on these contacts to cope. Similar results were recorded in a qualitative study of wellbeing in distance learning [37]. Here, study-related barriers and enablers were described as clear influencing factors. These were higher for more experienced students than for beginners.

With respect to the stress perception, students had much higher stress before the pandemic than during the online and the hybrid semesters. This is related to all the challenges and stresses related to their study life, knowledge acquirement, internships and examinations, etc. When these study stressors were stopped as the university was in lockdown during the first months of the pandemic, students’ stress perceptions declined. Medical students voluntarily helping in the CoronAid initiative had stress cores in the moderate range (18.1 ± 7.6) [26]. Their stress perception as volunteers in hospitals was nevertheless higher as compared to the preclinical students during their online semester as observed in this study (probably because they are faced with the concrete problems of the pandemic in the hospitals), while their stress perception was nevertheless lower than that of students before the pandemic. Of course, they were not confronted with and stressed by examinations in the university, and were able to practically help. It might be that the volunteers have experienced that their current knowledge and their practical skills developed during the medical studies are already important enough to help their future colleagues and current COVID-19 patients. This experience may have enabled them to cope better with their fears and worries, and thus with their stressors. Interestingly, the stress perception of students from the preclinical and clinical phases were similar, whether they are studying online or in the hybrid semester; and both had lower stress than before. This is where the opportunities for free time management and reduced commute times seem to pay off. Since students were well equipped and prepared for digital teaching, no new stressors were added [38].

Several studies found that female students were more sensitive towards stress than male students [4,11,26]. However, this was not generally approved in this study, as the stress perception declined in women and men without significant gender related differences. During the hybrid semester, male students’ stress perception was lower compared to female students, while it was similar in the online semester and prior to the pandemic. The reason for this observation is unclear. This could be attributed to differences in the socio-emotional assessment of stress under digital learning conditions in terms of a basic openness to digital learning and the digital competence that goes along with it [39]. Nevertheless, female students’ wellbeing decreased stronger compared to male students. It seems that both perceived a decrease of stress but also of their wellbeing. Yet, it is important to underline that students’ stress perception was not significantly associated with their declining wellbeing. Some may feel stressed, but this does not necessarily impair their life satisfaction, while other may have low wellbeing but they are not stressed. Instead, their stress perception negatively affects their life satisfaction and it is positively associated with cool down perceptions (which both are inversely associated). Their stress might be more triggered by the medical courses in-person, digital or hybrid (and related internships and examinations), rather than by the pandemic which affected their psychological wellbeing because of general social distancing, as it was observed in other cohorts, too [5,6]. Whether the significant decrease of general life satisfaction and work engagement in women during the hybrid semester, which was not significant in men, is really due to the education form or rather a hint of ‘hope fatigue’ because of the long phases of social restrictions, is unclear. Their work engagement refers to their medical studies, which would indicate that the effect should be attributed to the hybrid format. Further, there were no gender-related differences in the support satisfaction which remained high in all three waves, while the support received from their university in particular declined in the hybrid semester, and this was, however, perceived stronger by women.

How can these observations be explained? During the Summer Semester 2020 (W4) there was no vaccine available and thus students may have feared the risk of own infection, kept social distancing and were participating their online courses. In contrast, in the Winter Semester 2021/2022 (W7) the more infectious Delta virus variant was predominating, but students had the opportunity to receive a vaccination. During this phase of stronger increases of infected persons, the university courses were in hybrid format, online and in-person. Students’ fears of own infection might be lower (because of their vaccination), but social distancing was nevertheless required and this may have affected their psychological wellbeing. In fact, as shown in our study, students’ psychological wellbeing, life satisfaction and work engagement were still low also in the W7 sample, while the stress level slightly increased, probably because of the teaching format and protection requirements in the classes. Medical students’ life satisfaction was declining during their online and hybrid semesters of preclinical semester students, while there were no significant changes in students within their clinical semesters. The reason that preclinical semester students seem to be more challenged than clinical semester students might be explained by the new university and social environment with new duties and with examination tests and thus stress; yet, also the older students may be stressed due to emotional exhaustion because of the overload of knowledge and major examinations.

### 4.2. Wellbeing of Students

Are medical students really more anxious and depressed because of the pandemic? A systematic review by Lasheras et al. reported a 28% prevalence of anxiety in medical students during the pandemic, and this rate was “similar to that prior to the pandemic but correlates with several specific COVID-related stressors“ [40]. Findings from Nepal indicate that the rates of depression among medical students might be lower as compared to prior data [41]. In that study, the prevalence of depression was higher in students within their preclinical years as compared to students in their clinical years [41]. Yet, there are obvious cultural and geographic differences with higher prevalence in Western societies compared to Asian societies [13]. Students might be generally at larger risks for developing anxieties and emotional exhaustion [42], probably because they understand the underlying mechanisms of disease and related risks, but also because of an often (perfectionistic) personality [43], and academic pressure and financial difficulties [44]. However, particularly in the early phases of the pandemic, students’ insecurities how they may proceed with their medical studies, their clinical courses and finally their examinations might have been higher than in the later phases when universities have learned to deal with the restrictions and found ways to offer online courses and later on hybrid formats. This would fit to our findings of higher satisfaction with university support in the hybrid semester. Nevertheless, in our study, students from W4 and W7 perceived their stress similar, and also their wellbeing, life satisfaction and work engagement was similar—indicating that the format of education was not of crucial relevance for all outcome variables.

### 4.3. Predictors of Students Stress Perception, Psychological Wellbeing and Work Satisfaction

Our regression analyses to predict medical students’ stress perception, psychological wellbeing, and work engagement indicated that students stress perception can be explained only to some extent by their general dissatisfaction with their medical studies, whether they are studying in-person, online, or in a hybrid format. As their satisfaction with their studies was stably high also during the pandemic semester, their stress decline obviously has other reasons. Similarly, their work engagement (which did not change because of the pandemic related teaching formats) was predicted best by their satisfaction with the studies (which also remained stable for the three teaching formats) throughout all three waves of recruitment. To explain their wellbeing, satisfaction with their studies and the support received by their university was of some relevance in the regression model, but cannot explain too much of the wellbeing variance. Thus, the decline is related to the pandemic related general negative outcomes, not by the teaching formats. However, during their hybrid semester, their wellbeing could be explained much better by satisfaction with the medical studies (which was indeed high), by being in the clinical phase of the studies, by satisfaction with university support, by support received from by other students, and by being male. The hybrid semesters were obviously different in its outcomes on students, and it was applied in a later phase of the pandemic with high increases of infection incidences by the more aggressive Delta variant of the virus, but with the chance to be vaccinated. Thus, more chances for social contacts and less restricted teaching formats might explain some of the described observations, too.

### 4.4. Limitations

These data are from three cohorts of medical students recruited in a private university from one West German university, and thus, participants may not be representative for all medical universities in Germany. A special feature at Witten/Herdecke University is the individualized selection process that selects students according to their motivation for becoming a doctor, their ability to reflect, and their social commitment. This could result in this group being particularly adept at dealing with crises [45]. As several students have not responded to the survey, we cannot assess their stress perception and wellbeing. However, the proportion of women and men in all medical students of the Witten/Herdecke University (female 61%, male 39 %) is similar to the study responders (58% vs. 41%). Further, as at each semester new students enter in the sample and older students leave it, there is a large body of putatively similar students in the samples, but not the same students.

We relied on data from three recruitment waves prior and during the pandemic with their characteristic education characteristics as courses in-person, online, and hybrid. Nevertheless, the quality of life measures represent the impact of the pandemic rather than the education characteristics alone, which is nevertheless related to the pandemic.

There is a lack of qualitative analyses in these cohorts for a deeper understanding of the development of wellbeing and stress in the period of the pandemic and medical studies. This could be very useful, as—especially during the pandemic—there were multidimensional stress factors in private as well as professional life.

Of course, the general impact of technology-enhanced learning on stress and wellbeing must be considered in order to interpret the present results. Lister et al. differentiated skills and environmental circumstances [37]. Zupanic et al. [45] showed that students studying at the Witten/Herdecke University already used digital self-learning tools in face-to-face classes and that they were well equipped with hardware. However, high school graduates were not yet able to acquire these skills [46]. Interestingly, students who voluntarily choose distance learning prior to the pandemic showed high levels of wellbeing and self-management skills [47]. In these scenarios, students have voluntarily chosen distance learning. When using digital learning to supplement studies, a study by Gorbunovs et al. showed that a great deal of self-discipline is required to self-schedule learning [48]. Otherwise, an efficient learning setting easily becomes stressful. This may also explain the differences between experienced and inexperienced learners.

## 5. Conclusions

Although digital learning is reported to have beneficial effects, within the COVID-19 pandemic these formats were applied as a perforce substitute of regular in-person teaching formats which had to be stopped during the lockdowns. Our study indicates that the lockdowns affected students more than the teaching formats in “distance”, either online or in later hybrid formats. It has to be clearly underlined that medical students had no option to choose the teaching format best suited to them. Their stress perception decreased within the first phase of the pandemic and the online format, and started to rise with the switch to the hybrid format.

We have found a complex pattern of influences on their stress perception, psychological wellbeing and work engagement, as not all students seem to have found similar ways to cope with the outcomes of the pandemic and the course of their medical study. Particularly when there is no choice in teaching format, students have to take what is available on the one hand, and nevertheless require individual support when they cannot adjust to the difficult situation on the other hand.

Whether and to what extent universities can support all of their students in times of crisis are questions which cannot easily be answered. Nevertheless, universities need to develop student support structures at a school level, led by student ambassadors and graduates to motivate and emotionally support online learners. It was advised that medical schools facilitate help-seeking behaviors when their students feel stressed and anxious in order to avoid stigmatizing mental health problems [42]. It seems clear that students in their preclinical phase (lacking orientation, confirmation, and social contacts) require different support to students in their clinical phases (who are often also more stable in their personality). These are challenges for a medical education system that has to find ways to be prepared for future times of crisis and insecurity.

## Figures and Tables

**Table 1 ijerph-19-11098-t001:** Description of medical students of waves 1, 4 and 7.

KERRYPNX	All Students (W1 + W4 + W7)	Students W1	Students W4	Students W7	Significance *p*
**Number (n and %)**	1061 (100%)	327 (30.8%)	242 (22.8%)	490 (46.1%)	
**Participants (% of all medical students)**	53	59	36	62	
**Gender (%)**					n.s. ^1,2^
**female**	58.4	55.0	60.6	59.6	
**male**	41.1	45.0	39.4	39.4	
**diverse**	0.5	0.0	0.0	1.0	
**Mean age (years)**	24.4 ± 3.4	24.1 ± 3.3	24.4 ± 3.4	24.6 ± 3.6	n.s. ^1,2^
**Mean semester (semester)**	4.6 ± 3.0	4.8 ± 2.7	3.9 ± 3.2	4.8 ± 3.1	<0.001 ^1^; n.s. ^2^
**Phase of the study (%)**					<0.001 ^1^; n.s. ^2^
**Preclinical (1–4)**	58.1	47.3	73.5	53.5	
**Clinical (5–13)**	41.9	52.7	26.5	46.5	
**WHO-5 scores < 13 (%)**	28.1	14.9	34.2	35.3	<0.001 ^1^; n.s. ^2^
**Satisfaction with medical studies (BMLSS. item H3)**	4.5 ± 1.3	4.7 ± 1.0	4.5 ± 1.2	4.2 ± 1.5	n.s. ^1,2^
**Support Satisfaction (BMLSS-Support)**	72.8 ± 1.36	74.6 ± 1.3	74.3 ± 1.3	69.6 ± 1.5	n.s.^1,2^

^1^ comparison W1 vs. W4; ^2^ comparison W4 vs. W7; BMLSS: Brief Multidimensional Life Satisfaction Scale.

**Table 2 ijerph-19-11098-t002:** Indicators of stress and wellbeing in medical students of waves 1, 4 and 7 differentiated for the phase of their study.

	All Medical Students (Semester 1 to 13)					Preclinical Students 1 (Semester 1–4)					Clinical Students (Semester 5–13)				
	n	Mean	SD	Compared with	*p*	n	Mean	SD	Compared with	*p*	n	Mean	SD	Compared with	*p*
plStress perception (PSS)	W1: 321	23.3	5.15	W4	<0.001 **	W1: 136	22.0	4.93	W4	<0.001 **	W1: 60	24.5	5.07	W4	<0.001 **
	W4: 324	11.7	6.58	W7	<0.001 **	W4: 136	14.2	6.01	W7	<0.001 **	W4: 60	14.2	5.66	W7	n.s.
	W7: 465	17.0	3.73	W1	<0.001 **	W7: 136	17.1	3.39	W1	<0.001 **	W7: 60	16.4	4.63	W1	<0.001 **
Wellbeing (WHO-5)	W1: 324	17.2	4.21	W4	<0.001 **	W1: 138	17.8	3.74	W4	<0.001 **	W1: 62	16.2	4.55	W4	n.s.
	W4: 237	13.8	4.89	W7	n.s.	W4: 138	13.0	4.84	W7	n.s.	W4: 62	14.8	5.24	W7	n.s.
	W7: 472	13.8	5.14	W1	<0.001 **	W7: 138	13.7	4.96	W1	<0.001 **	W7: 62	14.9	5.24	W1	n.s.
Life satisfaction (BMLSS)	W1: 326	81.8	11.63	W4	0.047 *	W1: 127	84.0	10.04	W4	<0.001 **	W1: 51	79.2	11.69	W4	n.s.
	W4: 218	79.5	13.85	W7	0.036 *	W4: 127	77.5	14.00	W7	0.036 *	W4: 51	83.6	14.28	W7	n.s.
	W7: 457	75.6	15.01	W1	<0.001 **	W7: 127	74.7	13.92	W1	<0.001 **	W7: 51	78.9	13.61	W1	n.s.
Work engagement (UWES)	W1: 319	4.0	0.93	W4	n.s.	W1: 114	4.2	0.79	W4	n.s.	W1: 45	3.9	1.08	W4	n.s.
	W4: 202	3.8	0.94	W7	n.s.	W4: 114	3.9	0.89	W7	n.s.	W4: 45	3.6	1.11	W7	n.s.
	W7: 459	3.7	1.0	W1	0.014 *	W7: 114	3.7	0.99	W1	<0.001 **	W7: 45	3.7	0.79	W1	n.s.
Satisfaction university support (T2)	W1: 193	4.7	1.04	W4	n.s.	W1: 121	5.1	0.84	W4	<0.001 **	W1: 51	4.3	1.25	W4	n.s.
	W4: 193	4.5	1.23	W7	0.003 *	W4: 121	4.5	1.18	W7	n.s.	W4: 51	4.5	1.51	W7	n.s.
	W7: 193	4.1	1.50	W1	<0.001 **	W7: 121	4.5	1.13	W1	<0.001 **	W7: 51	3.9	1.59	W1	n.s.
Satisfaction student support (T1)	W1: 185	4.7	1.11	W4	n.s.	W1: 118	4.9	0.92	W4	0.005 *	W1: 49	4.3	1.41	W4	n.s.
	W4: 185	4.6	1.17	W7	n.s.	W4: 118	4.5	1.17	W7	n.s.	W4: 49	4.9	1.14	W7	n.s.
	W7: 185	4.6	1.33	W1	n.s.	W7: 118	4.6	1.35	W1	0.019 *	W7: 49	4.8	1.07	W1	n.s.
Satisfaction student cohesion (T3)	W1: 188	4.2	1.38	W4	n.s.	W1: 120	4.7	1.01	W4	0.020 *	W1: 48	3.9	1.33	W4	n.s.
	W4: 188	4.4	1.29	W7	n.s.	W4: 120	4.3	1.29	W7	n.s.	W4: 48	4.4	1.28	W7	n.s.
	W7: 188	4.2	1.44	W1	n.s.	W7: 120	4.4	1.36	W1	n.s.	W7: 48	4.3	1.14	W1	n.s.
Cool down (CDI)	W1: 307	16.3	6.18	W4	n.s.	W1: 71	15.6	5.73	W4	n.s.	W1: 37	17.0	5.24	W4	n.s.
	W4: 166	15.1	6.80	W7	n.s.	W4: 71	14.8	6.20	W7	n.s.	W4: 37	15.3	8.55	W7	n.s.
	W7: 390	15.9	8.07	W1	n.s.	W1: 71	14.4	7.55	W1	n.s.	W7: 37	14.9	6.23	W1	n.s.

* *p* < 0.05 (trend); ** *p* < 0.001 (significantly different); n.s.—statistically not significant (*p* > 0.05)

**Table 3 ijerph-19-11098-t003:** Indicators of stress and wellbeing in medical students of waves 1, 4 and 7 differentiated for gender.

	All Medical Students (Semester 1 to 13)					Female Students					Male Students				
	n	Mean	SD	Compared with	*p*	n	Mean	SD	Compared with	*p*	n	Mean	SD	Compared with	*p*
Stress perception (PSS)	W1: 321	23.3	5.15	W4	<0.001 **	W1: 130	23.9	4.97	W4	<0.001 **	W1: 83	22.3	5.13	W4	<0.001 **
	W4: 324	11.7	6.58	W7	<0.001 **	W4: 130	14.8	6.29	W7	<0.001 **	W4: 83	12.6	5.33	W7	<0.001 **
	W7: 465	17.0	3.73	W1	<0.001 **	W7: 130	17.6	3.50	W1	<0.001 **	W7: 83	15.8	4.40	W1	<0.001 **
Wellbeing (WHO-5)	W1: 324	17.2	4.21	W4	<0.001 **	W1: 132	17.1	4.23	W4	<0.001 **	W1: 84	16.6	4.07	W4	0.003 *
	W4: 237	13.8	4.89	W7	n.s.	W4: 132	13.6	4.86	W7	n.s.	W4: 84	14.1	4.83	W7	n.s.
	W7: 472	13.8	5.14	W1	<0.001 **	W7: 132	13.0	5.04	W1	<0.001 **	W7: 84	14.8	4.95	W1	0.037 *
Life satisfaction (BMLSS)	W1: 326	81.8	11.63	W4	0.047 *	W1: 122	81.6	11.75	W4	n.s.	W1: 72	81.6	11.16	W4	n.s.
	W4: 218	79.5	13.85	W7	0.036 *	W4: 122	78.6	14.69	W7	n.s.	W4: 72	80.9	11.88	W7	n.s.
	W7: 457	75.6	15.01	W1	<0.001 **	W7: 122	74.7	13.75	W1	<0.001 **	W7: 72	77.7	13.44	W1	n.s.
Satisfaction university support (T2)	W1: 193	4.7	1.04	W4	n.s.	W1: 118	4.7	1.05	W4	n.s.	W1: 70	4.7	1.06	W4	n.s.
	W4: 193	4.5	1.23	W7	0.003 *	W4: 118	4.5	1.20	W7	0.002 *	W4: 70	4.7	1.13	W7	n.s.
	W7: 193	4.1	1.50	W1	<0.001 **	W7: 118	3.9	1.54	W1	<0.001 **	W7: 70	4.5	1.21	W1	n.s.
Satisfaction student support (T1)	W1: 185	4.7	1.11	W4	n.s.	W1: 114	4.7	1.08	W4	n.s.	W1: 67	4.7	1.16	W4	n.s.
	W4: 185	4.6	1.17	W7	n.s.	W4: 114	4.5	1.21	W7	n.s.	W4: 67	4.9	1.06	W7	n.s.
	W7: 185	4.6	1.33	W1	n.s.	W7: 114	4.5	1.34	W1	n.s.	W7: 67	4.7	1.21	W1	n.s.
Satisfaction student cohesion (T3)	W1: 188	4.2	1.38	W4	n.s.	W1: 117	4.2	1.32	W4	n.s.	W1: 68	4.4	1.29	W4	n.s.
	W4: 188	4.4	1.29	W7	n.s.	W4: 117	4.2	1.32	W7	n.s.	W4: 68	4.6	1.25	W7	n.s.
	W7: 188	4.2	1.44	W1	n.s.	W7: 117	4.2	1.44	W1	n.s.	W7: 68	4.4	1.24	W1	n.s.
Work engagement (UWES)	W1: 319	4.0	0.93	W4	n.s.	W1: 107	4.0	0.88	W4	n.s.	W1: 67	3.9	1.00	W4	n.s.
	W4: 202	3.8	0.94	W7	n.s.	W4: 107	3.8	0.86	W7	n.s.	W4: 67	3.9	0.95	W7	n.s.
	W7: 459	3.7	1.0	W1	0.014 *	W7: 107	3.6	0.88	W1	0.004 *	W7: 67	3.8	0.96	W1	n.s.
Cool down (CDI)	W1: 307	16.3	6.18	W4	n.s.	W1: 76	16.1	6.45	W4	n.s.	W1: 44	16.0	5.00	W4	n.s.
	W4: 166	15.1	6.80	W7	n.s.	W4: 76	14.9	5.50	W7	n.s.	W4: 44	14.6	6.85	W7	n.s.
	W7: 390	15.9	8.07	W1	n.s.	W7: 76	14.8	7.57	W1	n.s.	W7: 44	17.0	8.99	W1	n.s.

* *p* < 0.05 (trend); ** *p* < 0.001 (significantly different); n.s.—statistically not significant (*p* > 0.05)

**Table 4 ijerph-19-11098-t004:** Correlations between indicators of stress and wellbeing within the whole sample of students.

	Semester	Stress Perception (PSS)	Wellbeing (WHO-5)	Life Satisfaction (BMLSS)	Support Satisfaction (BMLSS-Support)	Work Engagement (UWES)	Cool Down (CDI)
Semester	1						
Stress perception	0.110 **	1	0.039				
Wellbeing	−0.029	−0.039	1				
Life satisfaction	0.040	−0.235 **	0.309 **	1			
Support Satisfaction	−0.173 **	−0.164 **	0.192 **	**0.474 ****	1		
Work Engagement	−0.183 **	−0.145 **	0.239 **	**0.432 ****	**0.302 ****	1	
Cool Down	0.074	0.257 **	−0.096 **	−0.318 **	−0.270 **	−0.190 **	1

** *p* < 0.001 (Pearson correlation); moderate associations are highlighted (bold)

**Table 5 ijerph-19-11098-t005:** Predictors of students’ wellbeing (stepwise regression analyses).

		Beta	T	*p*	95% Confidence Interval	
					Lower Limit	Upper Limit
Dependent variable: WHO-5 in W1 F = 15.9. *p* < 0.0001; R^2^ = 0.10						
Model 2	(constant)		9.595	<0.0001	8.780	13.312
	Satisfaction with university’s support	0.181	2.792	0.006	0.193	1.115
	Satisfaction with studies	0.175	2.692	0.008	0.196	1.261
Dependent variable: WHO-5 in W4 F = 14.8. *p* < 0.0001; R^2^ = 0.17						
Model 3	(constant)		2.036	0.043	0.106	6.646
	Satisfaction with studies	0.249	3.441	0.001	0.421	1.552
	Satisfaction with university’s support	0.234	3.231	0.001	0.350	1.445
	Clinical phase of studies	0.151	2.328	0.021	0.246	2.972
Dependent variable: WHO-5 in W7 F = 47.5. *p* < 0.0001; R^2^ = 0.36						
Model 5	(constant)		−0.863	0.388	−3.350	1.305
	Satisfaction with studies	0.398	7.301	<0.0001	1.029	1.788
	Clinical phase of studies	0.197	4.713	<0.0001	1.175	2.855
	Satisfaction with university’s support	0.191	3.374	0.001	0.257	0.973
	Satisfaction with fellow students’ support	0.136	3.225	0.001	0.197	0.814
	Male gender	0.118	3.007	0.003	0.410	1.957

**Table 6 ijerph-19-11098-t006:** Predictors of students’ stress perception (stepwise regression analyses).

		Beta	T	*p*	95% Confidence Interval	
					Lower Limit	Upper Limit
Dependent variable: PSS in W1 F = 18.4. *p* < 0.0001; R^2^ = 0.16						
Model 3	(constant)		15.119	<0.0001	25.830	33.562
	Satisfaction with studies	−0.303	−5.446	<0.0001	−2.085	−0.978
	Clinical phase of studies	0.159	2.858	0.005	0.518	2.807
	Male gender	−0.109	−2.038	0.042	−2.252	−0.040
Dependent variable: PSS in W4 F = 17.1. *p* < 0.0001; R^2^ = 0.08						
Model 1	(constant)		13.805	<0.0001	18.101	24.135
	Satisfaction with studies	−0.284	−4.141	<0.0001	−1.977	−0.701
Dependent variable: PSS in W7 F = 24.5. *p* < 0.0001; R^2^ = 0.14						
Model 3	(constant)		30.669	<0.0001	21.571	24.526
	Satisfaction with studies	−0.278	−5.900	<0.0001	−0.955	−0.478
	Male gender	−0.185	−4.119	<0.0001	−1.989	−0.704
	Satisfaction with cohesion among students	−0.102	−2.175	0.030	−0.500	−0.025

**Table 7 ijerph-19-11098-t007:** Predictors of students’ work engagement (stepwise regression analyses).

		Beta	T	*p*	95% Confidence Interval	
					Lower Limit	Upper Limit
Dependent variable: UWES in W1 F = 78.8. *p* < 0.0001; R^2^ = 0.21						
Model 1	(constant)		9.065	<0.0001	1.589	2.471
	Satisfaction with studies	0.459	8.877	<0.0001	0.322	0.506
Dependent variable: UWES in W4 F = 24.3. *p* < 0.0001; R^2^ = 0.21						
Model 2	(constant)		9.090	<0.0001	2.121	3.297
	Satisfaction with studies	0.436	6.586	<0.0001	0.235	0.436
	Clinical phase of studies	−0.148	−2.242	0.026	−0.580	−0.037
Dependent variable: UWES in W7 F = 168.8. *p* < 0.0001; R^2^ = 0.28						
Model 1	(constant)		16.540	<0.0001	1.856	2.356
	Satisfaction with studies	0.533	12.993	<0.0001	0.317	0.430

## Data Availability

Due to data protection regulations, data are not publicity available.

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
