# Peer review of "Mental Stress in Medical Students during the Pandemic and Their Relation to Digital and Hybrid Semester—Cross-Sectional Data from Three Recruitment Waves in Germany"

_ijerph, 2022, doi:10.3390/ijerph191711098_

Round 1
Reviewer 1 Report
With great interest I have read the manuscript of Büssing et al. entitled “Mental stress in medical students during the pandemic and their relation to digital and hybrid semester – Cross-sectional data from three recruitment waves in Germany”.
In this study, the authors evaluated medical students´ stress perception, wellbeing, life and work satisfaction, and cool down reactions, and compared survey data of online and hybrid semesters with pre-pandemic education formats in presence.
The results supported that both students´ stress perception and their psychological wellbeing decreased during the pandemic semesters. Their satisfaction with the university support was lowest during the hybrid semesters.
Finally, the authors showed that the lockdowns affected students more than the teaching formats in “distance”. Their stress perception decreased within the first phase of the pandemic and the online format, and it started to raise with the switch to the hybrid format.
I believe that the manuscript can strengthen it by providing additional information on the following issue:
The title reflects well the contents of the paper.
The introductory section explains the framework and problems of the research, but the authors should take into consideration more recent cross-sectional study in Germany.
Methods, results, and data interpretation is partially addressed as above.
Participants
The study has not clear inclusion criteria.
The authors should provide information, how screening for exclusion of psychiatric diagnosis in the participants group was conducted.
What percentage of students in the study also were diagnosed with a psychiatric disorder?
The tables are easily readable and informative.
The conclusion is logically supported by the obtained results.
The present findings have important implications for clinical practice.
Author Response
Dear Reviewer
Thanks lot for your encouraging comments which helped us to improve the manuscript. - We hope that we have addressed all concerns adequately, so that the paper can be accepted now.
Comments and Suggestions for Authors
With great interest I have read the manuscript of Büssing et al. entitled “Mental stress in medical students during the pandemic and their relation to digital and hybrid semester – Cross-sectional data from three recruitment waves in Germany”.
>> Thanks a lot. We highly value this statement!
In this study, the authors evaluated medical students´ stress perception, wellbeing, life and work satisfaction, and cool down reactions, and compared survey data of online and hybrid semesters with pre-pandemic education formats in presence.
The results supported that both students´ stress perception and their psychological wellbeing decreased during the pandemic semesters. Their satisfaction with the university support was lowest during the hybrid semesters.
Finally, the authors showed that the lockdowns affected students more than the teaching formats in “distance”. Their stress perception decreased within the first phase of the pandemic and the online format, and it started to raise with the switch to the hybrid format.
I believe that the manuscript can strengthen it by providing additional information on the following issue:
The title reflects well the contents of the paper.
The introductory section explains the framework and problems of the research, but the authors should take into consideration more recent cross-sectional study in Germany.
>> Please let us know which studies you miss. We have re-scanned PubMed, but cannot find theses.
Methods, results, and data interpretation is partially addressed as above.
>> Is this a hint to change something? At least we are unclear what should be done.
Participants
The study has not clear inclusion criteria.
>> It does have: All students from the university studying medicine. However, we have underlined it more clearly (see yellow highlight).
The authors should provide information, how screening for exclusion of psychiatric diagnosis in the participants group was conducted.
>> We did not, as it was clearly communicated with the ethical board that we do not intend to pathologize stress perceptions and transient decreases of mood states. Thus, we are not allowed to ask for clinical diagnoses. Instead, we categorized persons “at risk” with low WHO-5 scores for this analysis. Moreover also Quek et al.(2019) argued that “administrators and leaders of medical schools should take the lead in destigmatizing mental illnesses and promoting help-seeking behaviors when students are stressed and anxious.”
To make our point clear, we added the following:
“Medical students from the Witten/Herdecke University, Germany (inclusion criterion), were invited to participate in this anonym survey via emails from the university´s vice president for Teaching and Learning to all medical students. To avoid pathologizing the students, we did not ask for (suspected) psychiatric diagnoses.”
What percentage of students in the study also were diagnosed with a psychiatric disorder?
>> As described above, we assume that our students were not treated for psychiatric disorders, and thus we did not ask for such diagnoses. Therefore, we cannot comment on this question.
The tables are easily readable and informative.
The conclusion is logically supported by the obtained results.
The present findings have important implications for clinical practice.
>> Thanks a lot
Reviewer 2 Report
Good day, the paper is highly important, as the issue of wellbeing is also present in more and more fields, including education. I have detected minor English grammar and spelling mistakes, that need to be addressed. Also, it would be wonderful if you provide the tables and any other significant data you have gathered, as well as any statistical formula that you have used. The tables are missing. Other than that, the paper is relevant and is useful!
Author Response
Good day, the paper is highly important, as the issue of wellbeing is also present in more and more fields, including education. I have detected minor English grammar and spelling mistakes, that need to be addressed.
>> Thanks for the hint. We checked it again and corrected some details. Hope we have found them all.
Also, it would be wonderful if you provide the tables and any other significant data you have gathered, as well as any statistical formula that you have used. The tables are missing.
>> All tables were uploaded together with the manuscript, and they were checked. However, indeed, they are not depicted in the current pdf. – We will now insert them directly in the manuscript to avoid such strange things.
Other than that, the paper is relevant and is useful!
>> Thanks a lot
Reviewer 3 Report
- Rewrite the conclusion section in two paragraphs
- Ref.no 42 is missing
- Some reverences are too old
- Complete the ref, with DOI
- Improve the discussion section with relevant references such is that published in IJERE Journal Published by IAES
Author Response
- Rewrite the conclusion section in two paragraphs
>> Thanks, we now have 3 paragraphs.
- Ref.no 42 is missing
>> Thanks, it was added again.
- Some reverences are too old
>> We used 48 references. Only a very few were no published in the 2020s. However, for essential information, such as reliance on health models or validation of measures, we cannot NOT cite them, i.e., Bakker´ Job-Demand-Model is a fundamental one, and it was published 2007. Cohen´s Stress scale validation was published 1983, indeed. The BMLLL validation was published
- Complete the ref, with DOI
>> Was added.
- Improve the discussion section with relevant references such is that published in IJERE Journal Published by IAES
>> The missing reference #42 is from the International Journal of Environmental Research and Public Health. Two further are already cited in the text. However, we are not sure why papers from the IAES International Journal of Artificial Intelligence should be cited.
Nevertheless, we additionally cited at the end of the conclusion from one IJERE paper: “Whether and to what extend universities can support all their students in times of crisis cannot easily be answered. Nevertheless, it seems that universities need to develop student support structures at school level led by student ambassadors and graduates to motivate and emotionally support online learners. It was advised that medical schools facilitate help-seeking behaviors when their students feel stressed and anxious to avoid stigmatizing mental health problems [42].”